# Association of eye strain with dry eye and retinal thickness

**Masahiko Ayaki** [1,2☯]*, **Manami Kuze**[3], **Kazuno Negishi**[1☯]

**1** Department of Ophthalmology, Keio University School of Medicine, Tokyo, Japan, **2** Otake Eye Clinic, Kanagawa, Japan, **3** Matsusaka General Hospital, Mie, Japan

☯ These authors contributed equally to this work.
* mayaki@olive.ocn.ne.jp

**Data Availability Statement:** All relevant data are within the paper and its Supporting Information files.

**Competing interests:** The authors declare no competing financial interests.

## Abstract

### Purpose

The purpose of this cohort study was to investigate the association between the prevalence of abnormal ocular examination results and the common visual symptoms of eye strain, blurred vision and photophobia.

### Methods

Consecutive first-visit outpatients with best-corrected visual acuity better than 20/30 in both eyes were enrolled and those with a history of intra-ocular lens implantation and glaucoma were excluded. Dry eye-related examinations and retinal thickness measurement were conducted. The odds ratio (OR) was calculated with logistic regression analyses of ocular data in relation to the presence of visual symptoms.

### Results

A total of 6078 patients (3920 women, mean age 49.0 ± 20.4 years) were analyzed. The prevalence of each symptom was 31.8% for eye strain, 22.5% for blurred vision and 16.0% for photophobia. A significant risk factor for eye strain was short tear break-up time (TBUT) (OR 1.88), superficial punctate keratitis (SPK) (OR 1.44), and thickness of ganglion cell complex (GCC) (OR 1.30). Risk factors for blurred vision were short TBUT (OR 1.85), SPK (OR 1.24) and GCC (OR 0.59). Risk factors for photophobia were short TBUT (OR 1.77) and SPK (OR 1.32). Schirmer test value, peripapillary nerve fiber layer thickness and full macular thickness were not associated with the tested symptoms.

### Conclusion

The current study successfully identified female gender, short TBUT, and SPK as significant risk factors for eye strain, blurred vision, and photophobia with considerable ORs.

**Abbreviations:** DE, dry eye; GCC, ganglion cell complex; ipRGC, intrinsically photosensitive retinal ganglion cell; NFL, retinal nerve fiber layer; OR, odds ratio; SPK, superficial punctate keratopathy; TBUT, tear break-up time.

# Introduction

Eye strain is a common ocular symptom in eye clinic patients with normal best-corrected visual acuity [1, 2]. It is a growing health problem that has been exacerbated during the COVID-19 pandemic due to increased screen time from telework and e-learning [3–8]. Digital eye strain due to virtual learning and other new technologies is an emerging social burden [4–8] and the prevalence of eye strain has been reported to be as high as 68.5% [5] and 81% [6] among visual terminal display users. Pavel et al. [6] reviewed and discussed numerous contributing factors for modern eye strain, including environmental and work factors, device-related factors, uncorrected refractive errors, ocular surface disorders, blink, and musculoskeletal modifications. Lotfy et al. [7] conducted a cross-sectional online survey of 412 participants and multiple logistic regression analysis revealed that studenthood and increased nighttime screen use were independent risk factors for digital eye strain by an odds ratio (OR) of 10.60 and 3.99, respectively. Presbyopia also seems to have developed earlier in life during the COVID-19 pandemic probably due to stress and increased digitalization leading to eye strain even in the younger population [9]. However, most reports describe review and survey results only and real examination data have not been sufficiently documented. Eye strain is a symptom with multiple presentations, including ocular pain, headache, heaviness, blurred vision, irritation, itching, lacrimation, and photophobia [5, 10]. It encompasses a broad spectrum of vision-related and nonvision-related pathologies, in addition to problems related to visual load, air conditioning, lighting, and ergonomics [4–8].

We have previously proposed a classification of eye strain (eye fatigue) and hypothesized that eye strain originates from both visual and nonvisual etiologies, with corneal hyperesthesia and photosensitivity playing major roles in nonvisual eye strain [11]. Neuropathic pain and intrinsically photosensitive retinal ganglion cells (ipRGCs) underlie this classification model, whereby corneal pain might be partly mediated by ipRGCs [12, 13]. IpRGCs are a subtype of RGCs that signal to the central biological clock to control circadian rhythm [14]. They are partly associated with photophobia and ocular pain in response to blue-light in the presence of melanopsin [14] and absence of rod and cone photoreceptor input [15]. IpRGCs project into the ciliary marginal zone [16] and iris [17], mediating innate light aversion to tissues in the anterior segments that receive trigeminal innervation [13]. Clinical investigations of glaucoma suggested ipRGCs are distributed in the retinal ganglion cell complex (GCC) [18–20] and thinning of the GCC may correspond to decreased ipRGC numbers or function. Although ophthalmological characteristics of ipRGCs are not determined clinically due to their low cell numbers, blurred vision may develop with a thinning of the GCC, as reported in glaucoma [21, 22]. Further, ipRGC-driven light sensitivity may be relieved with short wavelength light filtering eyewear to block blue-light and reduce light scattering [23–25]. Our previous investigation indicated GCC thickness was positively correlated with eye strain [11] and our hypothesis of ipRGC-associated eye strain is supported by numerous other investigations [26–31].

Dry eye (DE) is a major etiology of eye strain [10]. Unstable tear film and uneven corneal epithelial surface can disturb optical quality and induce higher order aberration and impaired functional visual acuity [32–35]. DE-associated visual symptoms can be treated with topical medications [36, 37] and they are effective in improving short tear break-up time (TBUT), superficial punctate keratopathy (SPK), and subjective symptoms including eye fatigue, blurred vision, photophobia and pain.

Eye strain has an unmet need in eye practice, mainly due to the difficulty in identifying causality, especially in cases without any detectable vision-related disorders. Few studies have evaluated eye strain in relation to objective examination data. This study aimed to investigate any association between abnormal ocular examination results and the common visual symptoms

of eye strain, blurred vision and photophobia, to complement our previous association of eye strain with retinal thickness in a larger sample size [11]. In addition, we attempted to identify risk factors for each symptom and explore relevant management for patients with normal vision complaining of visual symptoms.

## Methods

### Study design, patient recruitment and Institutional Review Board approval

This cohort study was a hospital-based, cross-sectional study conducted from April 2015 to March 2020. Outpatients were consecutively recruited to the study from Tsukuba Central Hospital (Ibaraki, Japan) from April 2015 to March 2020 and Otake Eye Clinic (Kanagawa, Japan) from December 2018 to March 2020. The data were accessed for research purposes at the Tsukuba Central Hospital from April 2015 to March 2020 and Otake Eye Clinic from December 2018 to March 2020. The institutional review boards and ethics committees of the Tsukuba Central Hospital (approved 12 December 2014, permission number 141201) and the Kanagawa Medical Association (approved 12 November 2018, permission number krec2059006) approved this study and it was conducted in accordance with the Declaration of Helsinki. The need for consent was waived by the institutional review boards. Minors were involved in this study and the need for consent from parents or guardians of the minors specifically waived. The institutional review board and ethics committee of Keio University School of Medicine approved this study (28 June 2021; approval number 20210080) to permit authorship for authors (KN and MA) with appointments at Keio University School of Medicine.

All the data used in this study, including the patient interviews, were collected as part of routine eye care examinations. The authors had access to information that could identify individual participants during and after data collection.

### Inclusion and exclusion criteria

Inclusion criteria were consecutive first-visit outpatients with best-corrected visual acuity better than 20/30 in both eyes. Exclusion criteria were a history of intra-ocular lens implantation or any ocular surgery within one month, glaucoma treated with medication, or any acute eye disease within one week. None of the patients had undergone any non-medical interventions, such as punctal plug insertion or punctal occlusion, or any surgical interventions. Consequently, a total of 6078 patients (3920 women, mean age 49.0 ± 20.4 years) were eligible for analysis (S1 Table). DE medication was used in 1578 patients (26.0%); hyaluronate in 782 (12.9%), diquafosol in 531 (8.7%), rebamipide in 84 (1.4%), and steroid in 179 (2.9%).

### Patient interviews for common eye symptoms

Participants were interviewed for the presence or absence (yes/no) of three common visual symptoms, namely eye strain, blurred vision and photophobia. These questions were selected from items on the Dry Eye–Related Quality-of-Life Score questionnaire [38] as the most prevalent symptoms seen in outpatient eye clinics at Keio University Hospital. The questionnaire comprised of three questions: "Do you feel eye strain?","Do you have blurred vision?", and"Do you feel sensitivity to light?".

### Ophthalmological examinations

Board-certified ophthalmologists with specialist expertise in retinal, glaucoma and corneal disorders submitted all participants to a routine examination that comprised visual acuity and intra-ocular pressure testing, biomicroscopy with vital corneal staining, and ophthalmoscopy.

Participants were also tested by the TBUT, Schirmer test with anesthesia, and vital corneal staining to detect SPK following standard procedures [1, 2].

Spectral domain optical coherence tomography (OCT) data were obtained using the RS 3000[R] (Nidek Co. Ltd., Aichi, Japan), and all OCT imaging was performed using the raster-scan protocol [11]. The thickness of the macular GCC (retinal nerve fiber layer [NFL] + ganglion cell layer + inner plexiform layer), peripapillary NFL, and the full retinal thickness in the central macular area were measured using built-in software.

## Statistical analysis

Data are presented as the mean ± SD or as percentages where appropriate. Schirmer's test, SPK, and TBUT test results from the more severe eye were used for analysis. The prevalence of short TBUT ($\leq 5$ s), SPK and low Schirmer test value ($\leq 5$ mm) was assessed in relation to the presence of visual symptoms. Using a logistic regression model, odds ratios (ORs) and 95% confidence intervals were calculated for DE-related signs and retinal thickness according to age and sex. All analyses were performed using StatFlex (Atech, Osaka, Japan), with $P < 0.05$ considered significant.

## Results

The demographics and examination results in relation to each symptom are shown in Table 1. The prevalence of each symptom was 31.8% for eye strain, 22.5% for blurred vision, and 16.0% for photophobia. The prevalence of each abnormal examination value was 53.0% for short TBUT, 23.5% for SPK and 6.8% for central involvement of SPK, and 28.8% for a low Schirmer test value. The mean age was significantly older in cases with blurred vision and photophobia. All symptoms were more prevalent in women than men. Short TBUT and SPK were more prevalent in individuals with all three symptoms compared with those without. Regarding retinal thickness, mean GCC was thicker in those with eye strain than in those without eye strain, and thinner in those with blurred vision than in those without blurred vision.

Table 1. Comparison of parameters between groups with and without visual symptoms.

| Parameters | Eye strain (-) | Eye strain (+) | P value† | Blurred vision (-) | Blurred vision (+) | P value† | Photophobia (-) | Photophobia (+) | P value† |
|---|---|---|---|---|---|---|---|---|---|
| No. of individuals | 4139 | 1929 | | 4699 | 1365 | | 5087 | 968 | |
| Age (years) | 48.7 ± 21.6 | 49.6 ± 17.6 | 0.12 | 47.2 ± 20.7 | 54.9 ± 18.0 | < 0.01 | 48.3 ± 20.4 | 52.2 ± 19.7 | < 0.01 |
| % male | 38.8 | 28.6 | < 0.01 | 36.9 | 30.6 | < 0.01 | 37.1 | 27.4 | < 0.01 |
| Dry eye-related signs* | | | | | | | | | |
| Short tear break-up time, n (%) | 1991 (48.1) | 1225 (63.5) | < 0.01 | 2335 (49.7) | 882 (64.6) | < 0.01 | 2579 (50.7) | 625 (64.6) | < 0.01 |
| Corneal epitheliopathy, n (%) | 882 (21.3) | 542 (28.1) | < 0.01 | 1057 (22.5) | 363 (26.6) | < 0.01 | 1150 (22.6) | 270 (27.9) | < 0.01 |
| Low Schirmer test value, n (%) | 161 (28.0) | 132 (29.9) | 0.51 | 189 (29.4) | 104 (27.9) | 0.90 | 225 (30.2) | 78 (25.4) | 0.13 |
| Retinal thickness | | | | | | | | | |
| Macular ganglion cell complex thickness (μm) | 92.4 ± 17.5 | 96.2 ± 18.1 | < 0.01 | 95.1 ± 17.3 | 90.7 ± 18.7 | < 0.01 | 94.0 ± 17.6 | 92.4 ± 18.4 | 0.12 |
| Peripapillary retinal nerve fiber layer thickness (μm) | 109.9 ± 18.4 | 111.5 ± 18.3 | 0.11 | 110.5 ± 18.6 | 110.3 ± 18.0 | 0.83 | 110.3 ± 18.3 | 10.8 ± 18.7 | 0.65 |
| Full retinal thickness of whole macula (μm) | 259.8 ± 30.0 | 260.8±29.4 | 0.64 | 260.8 ± 28.3 | 259.1 ± 32.2 | 0.39 | 60.4 ± 30.6 | 259.3 ± 26.7 | 0.61 |

† The tests for significance were *t* tests for continuous variables and Chi-square tests for categorical variables.

* Abnormal ocular surface parameters were defined as follows: tear break-up time $\leq 5$ s, Schirmer test value $\leq 5$ mm, the presence of corneal epitheliopathy (positive corneal staining).

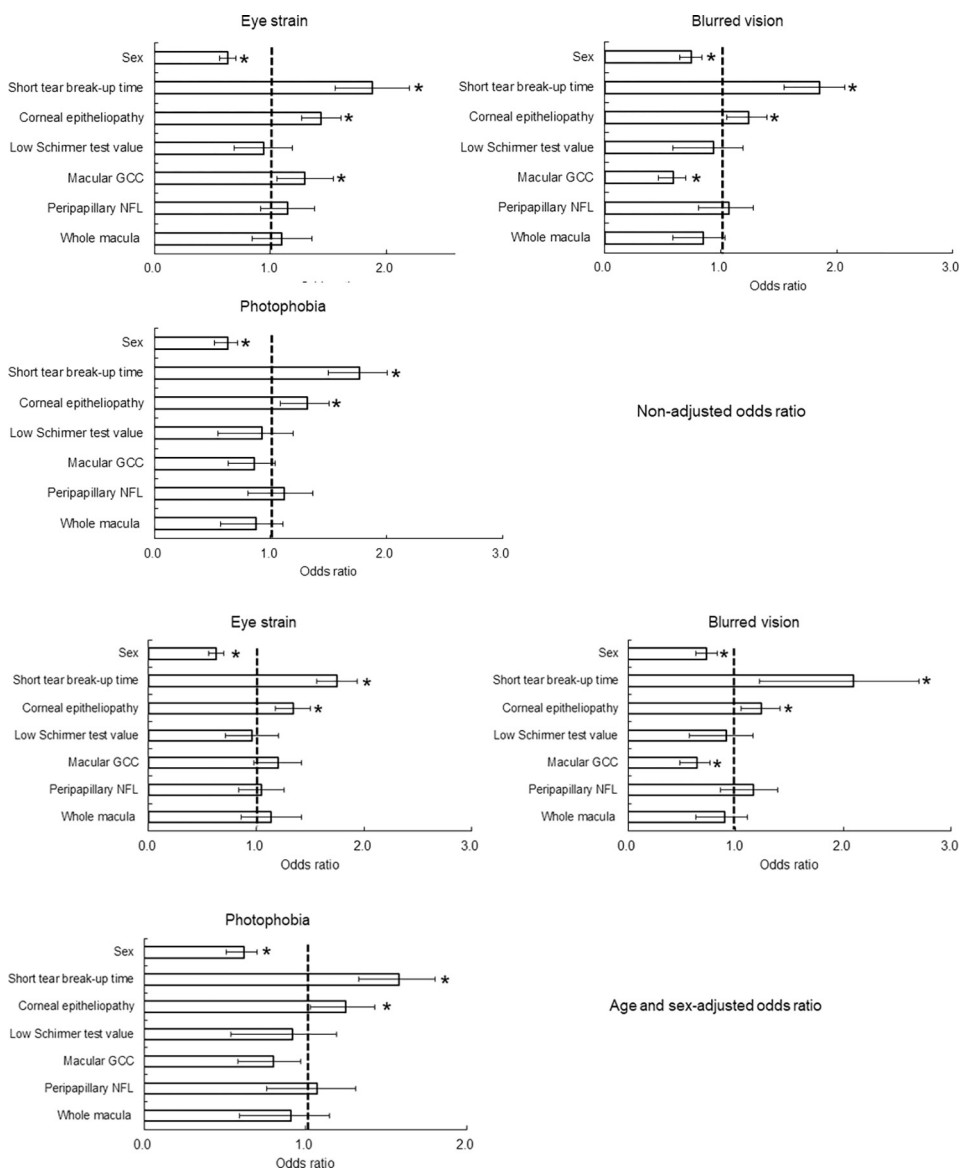

**Fig 1. Univariable logistic regression analyses of dry eye-related signs and retinal thickness in relation to the presence of eye strain, blurred vison, and photophobia.** Non-adjusted results are Fig 1A and age and sex adjusted results are Fig 1B. *P < 0.01. Odds ratios and 95% confidence intervals are indicated. Abnormal parameters were defined as follows: tear break-up time ≤ 5 s, Schirmer test value ≤ 5 mm, the presence of corneal epitheliopathy (positive corneal staining). Retinal thickness value was binarized by mean of the present cohort; thicker than mean = 1, thinner than mean = 0. Men = 1, women = 0. Abbreviations: GCC, ganglion cell complex thickness; NFL, retinal nerve fiber layer thickness; Whole macula, full thickness of whole macula.

Fig 1 and S2 Table show the univariable and age and sex adjusted logistic regression analyses of examination results in relation to the presence of the three visual symptoms. Individuals with short TBUT and SPK were more likely to present with eye strain, blurred vision and photophobia, while patients with thick GCC were more likely to present with eye strain, and patients with thin GCC were more likely to present with blurred vision. Schirmer test value, peripapillary NFL thickness and full macular thickness were not associated with the three symptoms tested.

## Discussion

The present study confirmed the results of recently published works [4–7], and linked underlying pathologies with patients' symptoms by demonstrating ORs calculated with objective data. The current study found short TBUT and SPK were significant risk factors for three visual symptoms with considerable ORs. This supports previous reports of both SPK and short TBUT being known causes of impaired visual function [33–35]. In the present cohort, short TBUT was observed as early disruption of tear film in the whole corneal area, whereas SPK was mostly located inferiorly, suggesting visual symptoms could develop even with local SPK not involving the central area. The current results confirm previous clinical findings that visual symptoms of DE could be treated with eyedrops by ameliorating TBUT and SPK [36, 37].

DE patients often complain of photophobia and it is generally believed to derive from light scattering on the ocular surface. However, in addition to developing from the optic media and retina, photophobia could also develop from the brain, systemic nervous system or psychological factors [39, 40]. Burstein et al. [39] introduced four definitions of photophobia: 1) abnormal sensitivity to light; 2) ocular discomfort, also termed photo-oculodynia; 3) exacerbation of headache by light; and 4) general aversion to light. Of these, ocular discomfort is often also applicable for eye strain-related photophobia and pain in DE and non-DE individuals since light inevitably enters the eyes while the eyes are open, especially when they are watching a digital display. The authors hypothesized that light may indirectly activate intraocular trigeminal nociceptors, that, in turn, activate second-order nociceptive neurons in the spinal trigeminal nucleus to induce ocular discomfort or pain in the eye. Other investigators have reported that a cervical sympathetic ganglion block reduced spontaneous pain and light sensitivity, and increased production of tears [40]. Based on these investigations, the trigeminal and sympathetic nervous systems may be involved in a subtype of eye strain.

The present study has further confirmed GCC thickening is a risk factor for eye strain. IpRGCs in the GCC may be associated with eye strain that causes discomfort or pain. This could be evoked through the trigeminal circuit after receiving short wavelength light emitted from lighting and displays [26–31]. Blue-light filtering eyewear may be effective for ocular symptoms related to ipRGC activation [23–25], however, it should be noted that sensitivity to light may depend on age, race and genetic factors [41, 42] and blue-light filtering does not work in certain populations. Thinning of the GCC was a risk factor for blurred vision and is a typical finding in Parkinson's disease [43] and vision-threatening diseases, such as glaucoma [21, 22], ischemic optic neuropathy [44] and optic neuritis [45].

Female gender was a significant risk factor for visual symptoms in the present study. Women complain of their symptoms more strongly than men, and genetic, hormonal, psychological and socio-cultural factors have been suggested to be associated with this observation [46]. One study also described girls may refrain from reporting their symptoms in adolescence [47]. As such, this gender difference should be recognized in the management of DE and other diseases. Age is also a considerable risk factor for eye strain. Younger individuals may suffer a more vigorous accommodative load and photo-oculodynia than the elderly due to better light transmission and retinal light sensitivity. Meanwhile, older individuals suffer DE-associated eye strain due to a susceptive ocular surface and decreased lacrimal function [48].

The current results support the contribution of symptom-oriented clinical reasoning combined with conventional statistics [49, 50], where practitioners first listen to the patient then correlate symptoms with examination results. In order to eliminate risk factors for eye strain, practitioners may prescribe DE eyedrops and consider recommending light blocking strategies through eyewear, shields and digital displays after excluding organic and functional disorders. Detailed examination may be required if a patient with a thin GCC suffers blurred vision;

namely, a visual field test, macular threshold and focal electro-retinogram. An OR could be a convenient threshold indicator to speculate the probability of symptoms in relation to particular objective data.

The present study has several strengths. First, the sample size is large enough to have the statistical power to calculate the OR in first-visit patients in a multiple logistic regression analysis. Second, the present analysis was performed on patients visiting before the COVID-19 pandemic. During the pandemic, many people changed lifestyle habits through a shift to telework and quarantining. This has likely impacted ocular, systemic and mental status and as such, means any generalizations from data during this period are unreliable. The current results may have sufficient generalizability since this is a two-center study with minimized bias.

The current study has some limitations. First, a functional visual acuity, contrast sensitivity, higher order aberration and accommodation test should have been performed to evaluate the severity of visual function in detail. Visual acuity corrected with patients' spectacles should also have been examined since inappropriate correction and excessive visual load are major causes of eye strain. Second, autonomic nerve activity could be further accessed to determine the neural contribution to visual symptoms since it is closely associated with eye strain-related symptoms including ocular discomfort, pain and photophobia. Eye pain could also be further evaluated with a validated questionnaire (such as the short-form McGill pain questionnaire) and esthesiometers. Third, the anatomy, physiology and function of human ipRGCs remain unclear. The measurement of ipRGC function with pupillary reflex or electroretinogram would be needed to determine the possibility of developing ocular symptoms. We must acknowledge the limitation of selection bias, as this study is clinic-based and people with any concerns may visit eye clinic.

In conclusion, the current study successfully identified female gender, short TBUT, and SPK as significant risk factors for eye strain, blurred vision, and photophobia with considerable ORs, and further confirmed GCC thickening is a risk factor for eye strain. Moreover, the present results contribute to understanding the association of visual symptoms with DE and short-wavelength light activating ipRGC-mediated ocular discomfort and pain and should help in the clinical reasoning and management of common visual symptoms in daily eye practice.

## Supporting information

**S1 Checklist. STROBE statement—checklist of items that should be included in reports of observational studies.**
(DOCX)

**S1 Table. The raw data of the subjects.**
(XLSX)

**S2 Table.**
(DOCX)

**S1 File.**
(DOCX)

**S2 File.**
(DOCX)

**S3 File.**
(PDF)

**S4 File.**
(JPG)

## Author Contributions

**Conceptualization:** Masahiko Ayaki.

**Data curation:** Masahiko Ayaki.

**Formal analysis:** Masahiko Ayaki.

**Methodology:** Masahiko Ayaki.

**Supervision:** Kazuno Negishi.

**Writing – original draft:** Masahiko Ayaki.

**Writing – review & editing:** Masahiko Ayaki, Manami Kuze, Kazuno Negishi.

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
