## [Decision Letter · Decision Letter 0]

6 Sep 2023

PONE-D-23-18713Association of eye strain with dry eye and retinal thicknessPLOS ONE

Dear Dr. Ayaki,

Thank you for submitting your manuscript to PLOS ONE. After careful consideration, we feel that it has merit but does not fully meet PLOS ONE’s publication criteria as it currently stands. Therefore, we invite you to submit a revised version of the manuscript that addresses the points raised during the review process.

We look forward to receiving your revised manuscript.

Kind regards,

Kofi Asiedu

Academic Editor

PLOS ONE

Journal Requirements:

2. Thank you for stating the following financial disclosure: "no"

3. Thank you for stating the following in your Competing Interests section: "no"

Reviewers' comments:

Reviewer's Responses to Questions

**Comments to the Author**

1. Is the manuscript technically sound, and do the data support the conclusions?

Reviewer #1: Partly

Reviewer #2: Yes

2. Has the statistical analysis been performed appropriately and rigorously? 

Reviewer #1: Yes

Reviewer #2: Yes

3. Have the authors made all data underlying the findings in their manuscript fully available?

Reviewer #1: Yes

Reviewer #2: Yes

4. Is the manuscript presented in an intelligible fashion and written in standard English?

Reviewer #1: No

Reviewer #2: Yes

5. Review Comments to the Author

Reviewer #1: Title: Association of eye strain with dry eye and retinal thickness

Eye strain, a prevalent issue among patients with normal vision, has surged during the COVID-19 pandemic due to increased screen usage in remote work and e-learning. It encompasses various symptoms like ocular pain, headache, and blurred vision, attributable to diverse factors such as visual load, lighting, and ergonomics. Notably, it is not always linked to visual tasks. A proposed classification model of eye strain suggests its origin from both visual and nonvisual factors, with corneal hyperesthesia and photosensitivity contributing significantly. Intrinsically-photosensitive retinal ganglion cells (ipRGCs), connected to circadian rhythm regulation and photophobia, might mediate corneal pain. Glaucoma research indicates these cells' distribution within the retina, potentially associated with decreased ipRGC function in thinning retinal areas. Dry eye emerges as a significant cause, disrupting visual quality through tear film instability. This study addresses the lack of clarity in eye strain causality by investigating connections between abnormal ocular exam findings and common symptoms like blurred vision and photophobia, complementing earlier research correlating eye strain with retinal thickness.

My comments:

Overall, this paper is well-written, demonstrating a significant effort. The study's aims and objectives are deserving of thorough investigation. While the study's foundation appears robust, the lack of clarity in the language employed poses challenges in comprehension. It would be advisable for the authors to engage with a writing coach or copyeditor to enhance the text's coherence and readability.

The study does not adequately connect the existing literature from prior research in this field to its aims and conclusions. It is recommended that the authors revise both the Introduction and Discussion sections to incorporate references to relevant literature, particularly focusing on recently published works.

Abstract:

1. The objectives outlined by the authors in the abstract does consistent with introduction are the titles of the study! I suggest having primary and secondary aims.

Abstract: “The purpose of this study was to investigate the association between the prevalence of abnormal ocular examination results and the common visual symptoms of eye strain, blurred vision and photophobia.”

Introduction:” this study aimed to investigate any association between abnormal ocular examination results and the common visual symptoms of eye strain, blurred vision and photophobia, to complement our previous association of eye strain with retinal thickness”

2. The conclusion lacks a comprehensive summary of the results.

Introduction

1. Lines 50-52 need references.

2. Line 49” The authors should use academic language instead of "It is an umbrella term"= it encompasses a broad spectrum of”.

Methods:

1. The demographic data should be incorporated within the Method section.

2. Additionally, it is advisable to provide more details regarding the yes/no questionnaire, including the total number of questions and their specific content.

3. Furthermore, it would be beneficial to provide references for the statement, "These symptoms were selected as the most prevalent in outpatients visiting the eye clinic of Keio University Hospital in 2012."

Results:

1. The author might consider utilizing alternative graphical representations to present the data, such as employing bar charts to facilitate a clearer comparison of the characteristics presented in Tables 2, 3, and 4. This approach could enhance the ease of understanding for the readers.

Conclusion

The discussion section should be expanded to create a more comprehensive link between the obtained results and the final conclusion.

Reviewer #2: This manuscript seeks to determine the risk factors/ association between retinal thickness, ocular surface diseases and measures (SPK, TBUT among others) and common visual symptoms such as eye strain, blurred vision and photophobia. This information is important, as understanding the effects of retinal thickness on the ocular surface examination and symptoms.

The manuscript is well written but I have few comments.

Title and abstract:

Overall, the title and abstract cover the main aspect of the work. However, the conclusion section of the abstract talks on the risk factors of eye strain only and ignored the risk factors of photophobia and blurred vision.

Introduction:

The background and information are relevant to the study.

Methods:

Overall, the methods are clear and can be replicated.

Line 100, the sentence does not read well. I would suggest the authors rephrase the sentence.

Line 114: The authors provided the year when patients' interviews for common eye symptoms was done (2012); however, the ethical approval was in 2021 for the Keio University and the earliest approval was in 2014. Kindly clarify.

Results;

The results are relevant. However, there was no information on the controls mentioned in the results section. I would suggest the authors provide details on the composition of the controls.

Discussion;

In the Discussion section, the results are discussed from various angles and placed into context without being over

interpreted. The indicated limitations of the study provide an opportunity to inform future research.

Conclusion;

I would suggest a stronger statement at the end.

6. PLOS authors have the option to publish the peer review history of their article (what does this mean?). If published, this will include your full peer review and any attached files.

Reviewer #1: **Yes: **Eman A. Alzghoul

Reviewer #2: No

---

## [Author Response · Author response to Decision Letter 0]

20 Sep 2023

Please find uploaded file of responses.

---

## [Decision Letter · Decision Letter 1]

10 Oct 2023

Association of eye strain with dry eye and retinal thickness

PONE-D-23-18713R1

Dear Dr. Ayaki,

We’re pleased to inform you that your manuscript has been judged scientifically suitable for publication and will be formally accepted for publication once it meets all outstanding technical requirements.

Kind regards,

Kofi Asiedu

Academic Editor

PLOS ONE

Additional Editor Comments (optional):

Reviewers' comments:

Reviewer's Responses to Questions

**Comments to the Author**

1. If the authors have adequately addressed your comments raised in a previous round of review and you feel that this manuscript is now acceptable for publication, you may indicate that here to bypass the “Comments to the Author” section, enter your conflict of interest statement in the “Confidential to Editor” section, and submit your "Accept" recommendation.

Reviewer #1: All comments have been addressed

Reviewer #2: All comments have been addressed

2. Is the manuscript technically sound, and do the data support the conclusions?

Reviewer #1: Yes

Reviewer #2: Yes

3. Has the statistical analysis been performed appropriately and rigorously? 

Reviewer #1: Yes

Reviewer #2: Yes

4. Have the authors made all data underlying the findings in their manuscript fully available?

Reviewer #1: Yes

Reviewer #2: Yes

5. Is the manuscript presented in an intelligible fashion and written in standard English?

Reviewer #1: Yes

Reviewer #2: Yes

6. Review Comments to the Author

Reviewer #1: Oveall, the authors has addresed all commnets .

I have one mionr commnet on abstract line 39: I suggest witing as following" The study found that female gender, short TBUT, and SPK are significant risk factors for eye strain, blurred vision, and photophobia with substantial odds ratios"

Reviewer #2: The authors have adequately addressed the comments. I would suggest the manuscript is accepted in its current form.

7. PLOS authors have the option to publish the peer review history of their article (what does this mean?). If published, this will include your full peer review and any attached files.

Reviewer #1: No

Reviewer #2: No

---

## [Editor Report · Acceptance letter]

13 Oct 2023

PONE-D-23-18713R1 

Association of eye strain with dry eye and retinal thickness 

Dear Dr. Ayaki:

I'm pleased to inform you that your manuscript has been deemed suitable for publication in PLOS ONE. Congratulations! Your manuscript is now with our production department. 

Kind regards, 

on behalf of

Dr. Kofi Asiedu 

Academic Editor

PLOS ONE